# Effect of Reactive Oxygen Species on the Endoplasmic Reticulum and Mitochondria during Intracellular Pathogen Infection of Mammalian Cells

**DOI:** 10.3390/antiox10060872

**Published:** 2021-05-28

**Authors:** Junghwan Lee, Chang-Hwa Song

**Affiliations:** 1Department of Medical Science, College of Medicine, Chungnam National University, 266 Munhwa-ro, Jung-gu, Daejeon 35015, Korea; asrai1509@gmail.com; 2Department of Microbiology, College of Medicine, Chungnam National University, 266 Munhwa-ro, Jung-gu, Daejeon 35015, Korea; 3Translational Immunology Institute, College of Medicine, Chungnam National University, 266 Munhwa-ro, Jung-gu, Daejeon 35015, Korea

**Keywords:** bacteria, ROS, mitochondria, ER stress, oxidative stress, pathogen, infection

## Abstract

Oxidative stress, particularly reactive oxygen species (ROS), are important for innate immunity against pathogens. ROS directly attack pathogens, regulate and amplify immune signals, induce autophagy and activate inflammation. In addition, production of ROS by pathogens affects the endoplasmic reticulum (ER) and mitochondria, leading to cell death. However, it is unclear how ROS regulate host defense mechanisms. This review outlines the role of ROS during intracellular pathogen infection, mechanisms of ROS production and regulation of host defense mechanisms by ROS. Finally, the interaction between microbial pathogen-induced ROS and the ER and mitochondria is described.

## 1. Introduction

Phagocytized pathogens are degraded in phagocytes, which is boosted by the so-called oxidative burst (or respiratory burst) [1]. The oxidative burst is an important component of immunity and removing microbes [1,2]. After infection, rapidly occurring oxidative burst is involved in the generation of nitric oxide and reactive oxygen species (ROS) [1,3]. Nitric oxide is generated by nitric oxide synthases (NOS), which convert L-arginine into L-citrulline and nitric oxide [4]. There are three NOS isoforms [5], one of which, inducible-NOS (iNOS), is induced in various cell types by cytokines and other agents in response to infection and inflammation [4,5]. Mitochondrial NOS-derived nitric oxide produces oxidative nitrogen oxide species, resulting in oxidative damage to mitochondria and apoptosis [6]. Several cellular signaling functions of nitric oxide are executed in mitochondria [6]. Nitric oxide inhibits activation of cytochrome c oxidase by competing with O_2_ for binding to cytochrome c oxidoreductase [6]. Nitric oxide reacts with complex III of the mitochondrial electron transport chain (ETC) to inhibit electron transfer and induce O_2_^•−^ generation [6]. Therefore, nitric oxide is involved in the regulation of host mitochondrial and cellular functions.

Oxidative stress is caused by an imbalance between the production and elimination of ROS [7]. Oxidative stress is linked to cardiovascular disease [8], cancer [9], neurodegenerative conditions [10] and respiratory diseases [11]. Activated inflammatory cell-induced ROS trigger irreparable tissue damage [12]. However, ROS are important for the treatment of various diseases. Cancer chemotherapy and radiation therapy are dependent on ROS-induced apoptosis [13]. ROS are also used as second messengers in multiple signal transduction pathways activated by proinflammatory cytokines [14]. In addition, ROS assist the immune system in eliminating pathogens [15]. ROS trigger immune defense systems such as production of proinflammatory cytokines, inflammation and autophagy [15,16]. Some pathogens modulate subcellular organelles, such as mitochondria and the endoplasmic reticulum (ER), which generate ROS [16,17]. Here, we review the relationship between pathogen infection and ROS.

## 2. Generation of ROS

### 2.1. ROS in Mitochondria

ROS are chemically reactive oxygen metabolites, produced as byproducts of aerobic metabolism in all living cells. Oxygen (O_2_) acts as a terminal electron acceptor in the mitochondrial ETC [18]. O_2_ accepts an electron, forming a superoxide anion radical (O_2_^•−^), causing ROS and oxidative stress [19]. Superoxide dismutase (SOD) catalyzes the conversion of O_2_^•−^ to the hydrogen peroxide (H_2_O_2_) and H_2_O_2_ performs as ROS [20]. ROS can be interconverted by enzymatic and nonenzymatic mechanisms. H_2_O_2_ is produced by natural or SOD-catalyzed conversion of O_2_^•−^ [21]. As such, mitochondrial ETC is a major source of intracellular oxidative stress [22].

The mitochondrial respiratory chain comprises four subunits (respiratory complexes I–IV) and two factors (cytochrome c, coenzyme Q10) in the mitochondrial intermembrane space [23] and mediates oxidative phosphorylation [24]. During ATP generation, electrons released from the mitochondrial ETC incompletely reduce O_2_ to superoxide [25]. Mn-SOD in the mitochondrial matrix or SOD in the intermembrane space convert superoxide into H_2_O_2_ [26]. In oxidative phosphorylation, 1–2% of O_2_ is incompletely reduced to superoxide [27], mainly in complexes I and III of the mitochondrial respiratory chain [26,28,29].

Complex I (NADH-ubiquinone oxidoreductase) uses coenzyme Q (fully oxidized form; ubiquinone) to transfer electrons from NADH to complex III (ubiquinol-cytochrome c oxidoreductase) [30]. The superoxide produced by complex I is secreted to the mitochondrial matrix [30]. Complex I generates superoxide by NADH-linked forward electron transport and succinate-linked reverse electron transport [27,31]. Mitochondrial superoxide production is triggered by an increase in the NADH/NAD^+^ ratio in the matrix [32]. Reverse electron transport from succinate to NAD^+^ also induces production of superoxide (Figure 1) [32].

Coenzyme Q is reduced to ubiquinol (reduced form of coenzyme Q) by receiving two electrons from mitochondrial complex I or II [30]. Complex III binds to ubiquinol in the intermembrane space and transfers electrons from ubiquinol to cytochrome c [30,33]. Cytochrome c accepts only one electron [30]. Rieske iron-sulfur protein (RISP) shifts an electron to cytochrome c_1_, forming ubisemiquinone (partially reduced form of coenzyme Q) [30]. Next, this electron is transferred to complex IV [30]. A second electron is transferred to cytochrome b_H_ from cytochrome b_L_ in complex III [30]. Superoxide is produced by electron of ubisemiquinone in complex III and released into the intermembrane space [30].

### 2.2. ROS in NADPH Oxidase

In immune cells, nicotinamide adenine dinucleotide phosphate (NADPH) oxidase (NOX) enables production of higher levels of ROS than mitochondria. NOX is a membrane-bound enzyme complex that produces ROS during pathogen invasion, as directed by the innate immune system [34]. NOX is a complex of gp91phox, p22phox, p40phox, p67phox, p47phox and Ras-related C3 botulinum toxin substrate (Rac) 1 or 2 [35]. gp91phox binds to p22phox to construct a transmembrane channel [36]. The regulatory proteins (p40phox, p67phox and p47phox) form a complex in the cytosol under normal conditions [37]. In stimulation conditions, p47phox is phosphorylated, causing localization of the regulatory protein complex at the gp91phox/p22phox complex [37]. Activated NOX transfers electrons to O_2_ on the opposite side of the cell membrane, yielding O_2_^•−^ [36]. Although not itself damaging, several by-products of O_2_^•−^, such as H_2_O_2_, HOCl^−^, OH^−^, OONO^−^ and NO_2_, are toxic (Figure 2) [38].

The gp91phox is also known as NOX2 [39]. The NOX family (homologs of gp91phox) consists of NOX1, NOX2, NOX3, NOX4, NOX5, dual oxidase (DUOX) 1 and DUOX2 [39]. The type of NOX family depends on the intracellular site and cell type [39]. NOX family members are implicated in the response to bacterial and viral infections and inflammatory signaling [39].

### 2.3. ROS in Endoplasmic Reticulum

The ER also generates ROS [40] during protein folding and formation of disulfide bonds [41]. The various ER oxidoreductases—such as protein disulfide isomerase (PDI), ERp29, ERp57 and ERp72—catalyze protein folding and oxidation [42]. ER oxidoreductases oxidize cysteine residues, generating disulfide bonds [42] and are reoxidized by ER oxidoreductin 1 (ERO1) [43]. ERO1 catalyzes protein disulfide bond formation using the oxidative power of molecular oxygen in ER [44], producing H_2_O_2_ [44]. The generation of disulfide bonds by ERO1 increases ROS generation [45]. PDI is necessary for formation of disulfide bonds in the ER [46]. During protein folding, PDI is reduced by the transfer of two electrons to a cysteine residue in the active site [47]. Reduced PDI causes incomplete reduction of oxygen, leading to formation of superoxide anion, which is converted to H_2_O_2_ or other ROS [41]. PDI is involved in all reactions that form disulfide bonds [48]. NOX4 is localized in the ER [46] and is activated by physical binding to PDI, resulting in NADPH-dependent production of H_2_O_2_ [46,49]. Excess ROS are eliminated by antioxidants such as glutathione peroxidases (Gpx) and peroxiredoxin (Prx) (Figure 3) [40,50].

ROS are required for normal cellular functions such as signal transduction, growth and proliferation [51,52]. Prolonged accumulation of ROS induces oxidative stress, leading to defects in genes and proteins [53]. Therefore, regulation of ROS is important for cell survival.

## 3. Role of ROS in Innate Immunity

### 3.1. ROS in Innate Immunity

Macrophages, neutrophils and dendritic cells recognize and phagocytose foreign substances and then activate acquired immunity by presenting a portion of foreign substances on their surface [54]. Pattern recognition receptors (PRRs) recognize pathogen-associated molecular patterns (PAMPs), such as glycans and glycoconjugates [55]. The activation of PRRs such as Toll-like receptors (TLRs), nucleotide-binding oligomerization domain (NOD)-like receptors (NLRs) and retinoic acid-inducible gene (RIG)-I-like receptors (RLRs), increases the expression of antimicrobial genes via the nuclear factor kappa B (NF-κB), mitogen-activated protein kinase (MAPK) and phosphatidylinositol 3-kinases (PI3K) pathways [56,57].

TLRs trigger proinflammatory cytokine and chemokine production by means of adaptor molecules such as myeloid differentiation primary response 88 (MYD88) and Toll/interleukin (IL)-1 receptor (TIR)-domain-containing adapter-inducing interferon-β (TRIF) [58]. TLRs are localized on the cell surface (TLR1, TLR2, TLR4, TLR5 and TLR6) or intracellularly [58]. Intracellular TLRs (TLR3, TLR7 and TLR9) are located in the endosomal membrane [59,60]. TLR2 recognizes bacterial, viral and fungal PAMPs, including peptidoglycans, lipoteichoic acid, lipoproteins, zymosan and mannan, via TLR1 or TLR6 [59]. TLR4 recognizes the lipopolysaccharides (LPS) of Gram-negative bacteria [59]. TLR5 recognizes the globular protein flagellin, a component of the bacterial flagellum [61]. TLR3 recognizes viral double-stranded RNA and small interfering RNA [62,63]. TLR7 recognizes viral single-stranded RNA [58]. TLR9 recognizes unmethylated CpG-DNA motifs in bacterial and viral DNA [61].

TLRs are closely associated with ROS production [64]. The production of TLR2 is increased by NOX4 in endothelial cells [65], increasing the response to TLR2 ligands in endothelial cells [65]. TLR4 activation by LPS causes recruitment of MYD88, leading to sequential activation of tumor necrosis factor (TNF) receptor-associated factor (TRAF) 6 and NF-κB [66]. The TIR domain of TLR4 interacts with the COOH terminus of NOX4, resulting in NF-κB activation and ROS production [67]. TLR5 activates NOX1 via NOX organizing protein 1 (NOXO1), a homologue of p47phox and p67phox [68,69,70]. Activation of TLR5 by recombinant flagellin from *Salmonella enteritidis* induced ROS production, which was amplified by NOXO1 overexpression [71]. Furthermore, TLR5 interacts with dual oxidase 2 (DUOX2), a member of the NOX family [72]. The DUOX2 C-terminal region is activated by the TIR domain of TLR5 in a calcium-dependent manner [72]. The activation of TLR7/TLR8 by agonists elevates ROS production via phosphorylation of p47phox, leading to neutrophil activation [73]. TLR9 activation by CpG-containing DNA induces ROS expression [74]. Therefore, TLRs modulate phosphorylation of members of the NOX family and ROS production.

Nucleotide-binding leucine-rich repeat receptors (NLRs) are intracellular PRRs that recognize cytosolic PAMPs [39]. NLRs typically have three domains—a central nucleotide-binding domain (NACHT) domain, C-terminal leucine-rich repeat (LRR) domain and N-terminal effector domain [75]. The N-terminal effector domain is subdivided into a caspase recruitment domain (CARD), pyrin domain (PYD) and acidic transactivating domain or baculovirus inhibitor repeats (BIRs) [75].

The activation signals for NOD1 and NOD2 have been identified. Activation of NOD1/NOD2 triggers RIP2 activation by NACHT domain oligomerization and CARD-CARD interaction [76]. Activated RIP2 induces production of proinflammatory cytokines by activating NF-κB [76]. NOD1 senses meso-diaminopimelic acid (meso-DAP) from Gram-negative bacteria [77]. NOD2 recognizes muramyl dipeptide (MDP), a peptidoglycan component of both Gram-positive and -negative bacteria [77]. DUOX2 produces ROS by MDP-mediated NOD2 activation [78].

The nucleotide-binding oligomerization domain (NLRP) subfamily of NLRs induces the inflammasome, thereby activating inflammation [79]. NLRP3 is a component of the NLRP3 inflammasome complex, which also comprises apoptosis-associated speck-like protein containing a CARD (ASC) and caspase-1 [80]. First, activated NLRP3 undergoes a homotypic NACHT domain interaction by self-oligomerization [80]. Next, the PYD domain of NLRP3 interacts with the PYD domain of adaptor ASC and another ASC domain, CARD, recruits the CARD domain of procaspase-1 [80]. The resulting NLRP3 inflammasome cleaves procaspase-1 to form caspase-1, leading to release of IL-1β [80]. NLRP3 is activated by various pathogens, intracellular DNA and RNA, mitochondrial DNA, ATP, Ca^2+^ and ROS [80]. ROS induce the NLRP3 inflammasome via the PI3K pathway, which activates Akt and extracellular signal-regulated kinase (ERK) 1/2 [81]. NLRP3 activators also induce production of ROS via the NLRP3 inflammasome [82]. Activation of the NLRP3 inflammasome requires TLR2/MyD88/NF-κB signaling and ROS/potassium efflux [83]. The activated NLRP3 inflammasome triggers IL-1β release [83]. In contrast, reduced production of ROS inhibits NLRP3 activation [84] and antioxidant treatment decreases inflammasome activation by suppressing NF-κB signaling [85]. Therefore, NLRs recognize and eliminate intracellular pathogens by inducing generation of ROS and so are important components of the innate immune system.

ROS trigger activation of the MAPK and NF-κB pathways [86,87,88]. Activation of immune signals such as JNK, ERK and NF-κB is required for secretion of proinflammatory cytokines and chemokines [86]. An elevated ROS level increases the expression of proinflammatory cytokines such as TNF, IL-1β and IL-6 [89,90]. *Pseudomonas* pyocyanin-induced ROS upregulated the mRNA levels of proinflammatory cytokines [91]. Similarly, antioxidants reduced p38 MAPK activation and production of monocyte chemoattractant protein 1 (MCP1) [92]. During *Helicobacter pylori* infection, inhibition of NOX by Korean red ginseng extract deactivates the Janus kinase (JAK) 2/signal transducer and activator of transcription (STAT) 3 pathway, downregulating MCP1 [93]. In immune cells, chemokine (C-X-C motif) ligand (CXCL) 1 activates the MAPK pathway, NF-κB pathway and production of ROS during bacterial infections [94]. MCP1 promotes the release of lysosomal enzymes [95,96] and induces p47phox expression in macrophages and neutrophils, leading to ROS production [97]. Therefore, chemokines and cytokines are closely related to ROS production by members of the NOX family.

Enhanced PRR signaling by ROS contributes to pathogen elimination. Macrophages recognize bacterial ligands via PRRs and produce mitochondrial ROS (mtROS) [98]. Innate immune cells (macrophages and neutrophils) release ROS to degrade the pathogen [98]. ROS are important intracellular mediators of the antimicrobial response and tune the inflammatory response [98].

### 3.2. ROS in Bacterial Pathogenesis

ROS kill pathogens directly by causing oxidative damage or indirectly by stimulating nonoxidative mechanisms [16]. Mitochondrial-targeted catalase expression inhibited the killing of *Salmonella typhimurium*, demonstrating that mitochondrial ROS can be bactericidal [99]. In addition, stimulation of TLR9 by CpG-containing DNA and subsequent ROS production enhanced killing of *Staphylococcus aureus* in osteoblasts [100]. TLRs affect accumulation of mtROS by inducing mitochondria-derived vesicles in *S. aureus*-infected macrophages [101]. ER stress increases production of mitochondrial vesicles, which contain SOD [101]. *S. aureus* infection triggers production of mitochondrial vesicles in a Parkin-dependent manner, leading to mtROS accumulation in bacteria-containing phagosomes [101]. Depletion of SOD2 reduces mitochondrial H_2_O_2_ production and suppresses the elimination of bacteria [101].

The interaction of mtROS and TLR signaling influences inflammation [102]. mtROS are produced by increased activity of mitochondrial complex II as a result of activation of the Src-type tyrosine kinase, Fgr, which is triggered by NOX-derived ROS in macrophages infected with *Escherichia coli* [102,103]. This process requires TLR signaling and the NLRP3 inflammasome, which are, thus, critical for bacterial survival [102].

Production of ROS is involved in various physiologic responses, such as transcriptional activation, proliferation and apoptosis [104]. NOX is important for innate immunity [104]. Patients with chronic granulomatous disease (CGD) cannot generate ROS because of a NOX genetic defect [105], suppressing the production of ROS precursors [106]. CGD mainly affects the lungs, lymph nodes, skin and liver and enhances susceptibility to bacterial and fungal infections, leading to pneumonia, abscesses, suppurative arthritis and osteomyelitis [106]. Macrophages activated by interferon γ (IFN-γ) retained *Listeria monocytogenes* within phagosomes and inhibition of ROS enables its escape from phagosomes to the cytosol [107]. In addition, *L. monocytogenes* more easily escapes phagosomes in macrophages from NOX subunit gp91phox-deficient mice [107]. The *Francisella tularensis* survival ratio was increased in lungs and spleens from gp91phox^−/−^ compared to wild-type mice [108]. *F. tularensis* impairs neutrophil activation and disrupts assembly of NOX in phagosome membranes, inhibiting ROS production [108].

Recognition of bacteria by macrophages suppresses the assembly of ETC complex I and ETC complex I-containing super-complexes [102]. Simultaneously, ETC complex I and ETC complex II increase mitochondrial respiration [102]. This process is mediated by phagosomal NOX and the ROS-dependent tyrosine kinase Fgr and leads to induction of ROS [102]. An increased ROS level enhances accumulation of the proinflammatory cytokine IL-1β, activating an inflammatory response and enhancing pathogen killing [102]. Furthermore, generation of ROS by proinflammatory cytokines such as IFN-γ and TNF is important for eliminating pathogens in macrophages [12,109,110]. However, production of ROS can damage host cells. Indeed, excessive TNF promotes production of ROS and induces mitochondrial Ca^2+^ overload in *Mycobacterium tuberculosis* (Mtb)-infected macrophages, causing necrosis [109]. Therefore, infected immune cells induce ROS production to inhibit bacteria and induce an inflammatory response.

### 3.3. ROS in Viral Pathogenesis

Virus-induced ROS trigger the intrinsic apoptosis pathway mediated by Cyt c and caspase-9 [88]. Apoptosis is important for controlling intracellular pathogens [111]. Infection by viruses, such as Japanese encephalitis virus (JEV) activates the NF-κB pathway by inducing ROS generation, leading to upregulation of antiviral genes in lymphocytes [112,113]. Survival of herpes simplex virus 1 (HSV1) was reduced by release of ROS in neutrophils [114]. Thus, ROS generation by NOX and mitochondria is critical for suppression of viruses.

Virus-mediated ROS activate the inflammasome [115]. The NLRP3 inflammasome is activated by RNA viruses, DNA viruses, bacterial RNA, ROS, ATP, mitochondrial DNA and K^+^ efflux [116,117]. Adenoviral DNA triggers assembly of the NLRP3 inflammasome, activation of caspase-1 and secretion of IL-1β [118]. Activated caspase-1 regulates production of the pro-pyroptotic factor gasdermin D (GSDMD) [119]. GSDMD induces pyroptosis by forming pores in the membrane of virus-infected cells [120]. Pyroptosis increases inflammation and release of IL-1β [120]. Secreted IL-1β induces neutrophil recruitment at inflammation sites to kill viruses [121]. ROS-induced NLRP3 inflammasome activation is responsible for host antiviral activity.

Human cytomegalovirus (HCMV) increases the levels of antioxidant enzymes, such as SOD, glutathione peroxidase 1(GPx1) and glutamate cysteine ligase (GCL), leading to suppression of ROS [122]. However, HCMV activates ROS production via NF-κB in host cells [123]. HCMV-induced ROS activate viral major immediate promoter (MIEP) via the immediate-early protein IE72 [123]. During viral infection, induction of ROS is required for production of IE72 and replication of HCMV [123]. The TAT protein of human immunodeficiency virus (HIV) inhibits the antioxidant enzyme MnSOD, inducing ROS production [124]. Viruses disrupt the antioxidant system, leading to excess ROS, triggering inflammation and damage, thus promoting viral replication [125].

Hepatitis C virus (HCV) is the etiologic agent of acute and chronic hepatitis, fibrosis, cirrhosis and primary hepatic malignancy [126]. The level of ROS is elevated in liver tissue of patients with chronic HCV infection [127]. HCV core protein directly interacts with mitochondria, causing ROS generation during chronic hepatitis C of the mouse liver [128]. In addition, HCV non-structural protein 3 (NS3) increases the intracellular Ca^2+^ level and generates ROS via phosphorylation of p47phox in human monocytes [129]. Ca^2+^ signaling is essential for ROS production—a calcium channel inhibitor (lanthanum chloride) reduced ROS production in HCV-infected cells [129]. During HCV infection, NOX4 triggers induction of superoxide and H_2_O_2_ in hepatocytes [130,131]. The NOX4 mRNA level was increased in HCV-infected human hepatocytes (Huh-7) [130]. HCV cDNA elevated the mRNA and protein levels of NOX4 [131]. Therefore, NOX4 is a major source of ROS production in HCV-infected host cells [130,131].

Influenza virus infections induce acute respiratory diseases, eventually triggering cell death in the respiratory system [132], generating a cytokine storm, which damages local tissue and causes systemic sequelae [132,133]. ROS induce excessive TNF-α, IFN-γ, IL-6 and inducible nitric oxide synthase (iNOS) production during influenza virus infection, causing excessive immune responses and inflammation [134]. However, ROS suppress influenza virus by killing infected cells [135]. Indeed, NOX genetic deficiency and ROS inhibitors induce viral replication during pulmonary influenza virus infection [136].

Severe acute respiratory syndrome coronavirus-2 (SARS-CoV-2) induces sepsis in severe cases [137]. SARS-CoV-2 sepsis induces hypoxia, which can promote mitochondrial dysfunction leading to production of superoxide, H_2_O_2_ and other ROS [138,139,140]. In addition, virus-induced ROS damage the erythrocyte membrane [141] and damaged erythrocytes are phagocytosed by macrophages and neutrophils [138], further increasing ROS production [138]. The interplay between excess ROS and cytokines induces a cytokine storm and oxidative stress production, leading to death from sepsis and shock [138]. Cytokine storm causes acute respiratory distress syndrome and multiple organ dysfunction in patients with SARS-CoV-2 [142,143,144,145,146]. Therefore, regulation of ROS could have therapeutic potential for SARS-CoV-2 infection.

In summary, several viruses use ROS for survival, whereas others are suppressed by ROS. Thus, ROS production can be beneficial or harmful during viral infection, depending on the virus in question.

## 4. Oxidative Stress-Mediated ER Stress

The ER is a specialized organelle that contributes to maintenance of cell homeostasis by regulating protein biosynthesis and folding, lipid metabolism and Ca^2+^ homeostasis [147]. Impairment of ER function by physiological or pathological factors is termed ER stress [148]. During ER stress, the protein processing ability of ER is weakened, causing accumulation of misfolded proteins [149]. Accumulation of misfolded proteins severely disrupts maintenance of ER homeostasis and initiates unfolded protein response (UPR) [149,150]. The UPR is a mechanism for removing misfolded or unfolded proteins and can activate immune signaling, inflammation and apoptosis [151]. In mammalian cells, activation of the UPR induces ER chaperone binding immunoglobulin protein (BiP), which activates inositol-requiring enzyme 1 (IRE1), activating transcription factor (ATF) 6 and protein kinase R (PKR)-like ER kinase (PERK) [151,152]. IRE1 and PERK form homodimers or oligomers and autophosphorylate [153]. ATF6 is transported to the Golgi apparatus, where it is processed and activated [154].

Activation of UPR signaling elevates production of ROS during acute or chronic ER stress [155]. The UPR pathway is activated during ER stress by oxidants, including ROS, peroxides and metal ions [156]. In macrophages, antioxidants reduce the UPR response, suggesting that ER stress induction is dependent on oxidative stress. [157]. Furthermore, accumulated misfolded proteins induce ER stress, leading to ROS production [45,158,159]. Therefore, ROS induce ER stress and ER stress increases ROS levels. Prolonged ER stress causes failure of ER homeostasis, increasing apoptosis [160]. The eukaryotic initiation factor (eIF) 2-ATF4 axis, downstream of PERK, activates transcription factor C/EBP homologous protein (CHOP), thereby inducing the synthesis of pro-apoptotic proteins [161]. In addition, CHOP targets Ero1 (a producer of ROS), leading to ROS-mediated apoptosis [162].

As well as the ER, some NOX4 is present in the mitochondria, nucleus and cytoplasm [163]. NOX4 activity requires its interaction with p22phox [39]. Activated NOX4 complex generates superoxide anion using NADH or NADPH as an electron donor [164]. In peripheral vasculature cells, the UPR response induces ROS production via NOX4 activation [165]. The ER stress induced by tunicamycin and 7-ketocholesterol leads to increased NOX4 mRNA and protein levels, resulting in increased ROS production and apoptosis [155,157]. In addition, the p22phox site of NOX4 physically binds PDI (an ER protein-folding enzyme), leading to ROS production [49]. The interaction of p22phox and PDI occurs in infected macrophages [166]. In macrophages, PDI regulates ROS production [166]. Following activation of the UPR response by ER stress, NOX4 and PDI regulate cell fate by modulating ROS production. Therefore, the pathogen-induced UPR response triggers ROS-mediated apoptosis, thus suppressing pathogens.

Oxidative stress releases calcium from the ER into the cytosol in the early stage of ER stress [167]. Cytosolic calcium is absorbed by mitochondria through voltage-dependent anion channels (VDAC) [167]. Mitochondrial calcium overload induces production of mtROS and causes mitochondrial damage, amplifying mtROS production [168]. ER stress-mediated Ca^2+^ release occurs via opening of inositol triphosphate receptors (IP3R) [169]. Elevated mtROS induce Ca^2+^ release via IP3R, boosting calcium release from the ER and mtROS production [170,171]. In addition, calcium stimulates NOS, which suppresses ETC complex IV, inducing mtROS generation [46]. During ER stress, calcium release affects mitochondria and modulates mtROS production. Therefore, regulation of ER stress and calcium release is critical for ROS generation.

Oxidative stress is closely related to ER stress during pathogen infection and triggers production of proinflammatory cytokines, generation of ROS, induction of inflammasome, autophagy and apoptosis (Figure 4) [172,173,174]. HCV increases ROS-induced ER stress, resulting in activation of proinflammatory signals, autophagy and apoptosis [175,176,177]. ROS induce ER stress and the ASK1/ERK/p38 MAPK pathway, which trigger apoptosis during Japanese encephalitis virus infection of human promonocytes [178]. In the case of bovine viral diarrhea virus, induced ER stress activates ROS-induced apoptosis [179]. *Streptococcus pneumoniae* induces cytosolic ROS accumulation, triggering ER stress and elevating the production of proinflammatory cytokines [174]. ER stress-induced ROS are also required to kill methicillin-resistant *S. aureus* [180]. Toxins such as *Brucella melitensis* toxin TcpB, *Brucella abortus* toxin VceC, Shiga toxin and listeriolysin O are involved in ER stress and are critical for the pathogenesis of infectious diseases [181,182,183,184].

The 6 kDa early secretory antigenic target (ESAT-6) produced by Mtb induces an ER stress response, release of Ca^2+^ from the ER into the cytosol and accumulation of ROS, causing apoptosis [185]. Mtb 38 kDa antigen (38 kDa Ag) triggers the activation of MAPK signaling cascades (JNK, ERK and p38 MAPK) following stimulation of TRL 2/4 in macrophages [186]. Activated MAPK increases the secretion of proinflammatory cytokines (MCP-1, TNF-α and IL-6) and MCP-1 initiates production of MCP-1 inducible protein (MCPIP) in 38 kDa Ag-treated macrophages [186]. MCPIP is linked to induction of p47phox, ROS and ER stress [187]. In MCPIP-deficient macrophages, ROS generation and ER stress-mediated apoptosis are reduced even after stimulation with 38 kDa Ag, suggesting that MCPIP suppresses intracellular mycobacteria [186]. In addition, live Mtb more strongly activates production of ROS, NO and CHOP than heat-killed Mtb in macrophages [188]. Prostate apoptosis response-4 (Par-4; a tumor suppressor protein) was induced by Mtb-mediated ROS in macrophages [189]. The produced ROS increase apoptosis by inducing formation of the Par-4-BiP complex, leading to suppression of intracellular growth of Mtb [189]. Mtb-mediated ROS activate production of calreticulin (CRT; a calcium-binding chaperone protein), increasing ER stress-mediated apoptosis of macrophages [190]. Therefore, ROS-induced ER stress is critical for inhibition of Mtb (Figure 5).

*Mycobacterium kansasii*-infected macrophages show elevated intracellular ROS production, leading to ER stress-mediated apoptosis [191]. *M. fortuitum*-induced ROS trigger production of TNF-α via the NF-κB/JNK pathway, leading to ER stress-mediated apoptosis of macrophages [192]. Avirulent *Mycobacterium smegmatis* induces high levels of ROS production, ER stress sensor molecules and secretion of proinflammatory cytokines in macrophages [193]. Phagocytosis of *M. smegmatis* triggers production of ROS and proinflammatory cytokines via the TLR signaling pathway in macrophages [193]. *Mycobacterium avium* also initiates ROS-mediated ER stress in macrophages [194]. *M. avium*-induced ER stress activates the regulated IRE1-dependent decay (RIDD) pathway via IRE1α RNase [194]. The RIDD pathway increases ER stress by degrading the mRNAs of ER chaperone proteins and anti-apoptotic microRNAs, initiating apoptosis [195,196]. The ROS scavenger NAC reduces IRE1α and apoptosis in macrophages infected with *M. avium* [194]. Therefore, not only Mtb but also various non-tuberculosis mycobacteria (NTM) induce ROS synthesis and ER stress in macrophages.

During infection with bacteria or viruses, ROS regulate ER stress induction and activate host defense mechanisms, such as the NF-κB pathway, MAPK signaling, proinflammatory cytokines and apoptosis. Therefore, ROS-mediated ER stress is critical for pathogen elimination.

## 5. Alteration of Mitochondrial Dynamics by Oxidative Stress

Mitochondria are dynamic organelles involved in metabolism, differentiation and cell death [197]. Mitochondria produce ATP via the electron transport chain [198]. Mitochondrial dynamics are important for ROS generation and are maintained by mitochondrial fusion and fission [197,198]. Fusion and fission of mitochondria happen constantly, regulating their size and subcellular spread and revealing their functional state [199]. Key players in the fusion process are the outer mitochondrial membrane GTPases mitofusin (MFN)1 and MFN2 and the inner membrane GTPase optic atrophy 1 (Opa1) [200,201]. MFN1 and MFN2 induce fusion of the outer mitochondrial membrane (OMM) [202]. MFNs oligomerize in mitochondria, bringing mitochondrial membranes closer and inducing OMM fusion [202]. MFN1 has a greater ability to tether mitochondria because of its higher GTPase activity than MFN2 [203]. MFN2 is required for mitochondrial-associated membrane (MAM) formation [203]. After OMM fusion, OPA1 induces fusion of the inner mitochondrial membrane (IMM) [204]. Mitochondrial fission requires mitochondrial fission 1 protein (Fis1, located in the OMM [205]) and GTPase dynamin-related protein 1 (DRP1) [206]. Fis1 recruits DRP1 from the cytosol, which coalesces into foci at mitochondrial cleavage sites [205]. The GTPase activity of DRP1 is elevated by CDK-1 dependent phosphorylation [207]. MARCH-V, an OMM transmembrane protein, triggers ubiquitination of DRP1 and reduces mitochondrial fission [208]. Fragmentation of mitochondrial networks is increased by a reduction in the levels of fusion proteins or an increase in those of fission proteins [209]. Mitochondrial fragmentation is linked to mitochondrial dysfunction, including loss of mitochondrial membrane potential (MMP), decreased oxidative phosphorylation (OXPHOS), metabolic shift towards glycolysis and increased mitochondrial ROS formation [198]. Mitochondrial fragmentation enhances ROS formation, causing a deterioration in mitochondrial health and further exacerbating oxidative stress [210]. Mitochondrial fragmentation-induced mitochondrial dysfunction can promote selective autophagy, such as mitophagy, or trigger apoptosis under severe oxidative stress conditions [211,212].

Mitochondria are also targets of pathogens [213,214]. Macrophage polarization is regulated by mtROS, the generation of which is modulated by mitochondrial dynamics [204,215]. Therefore, alteration of mitochondrial dynamics is important for suppression of pathogens. HCV-induced mitochondrial dysfunction triggers mtROS production [216]. Indeed, HCV non-structural protein 5A (NS5A) induces ROS, leading to mitochondrial fragmentation and loss of MMP. HCMV, influenza virus and measles virus disrupt the mitochondrial network [217,218,219,220]. In addition, Venezuelan equine encephalitis virus damages mitochondrial function, leading to mitochondrial fragmentation and ROS generation [221].

*S. pneumoniae* increases mitochondrial fragmentation-induced mtROS, leading to suppression of bacteria [222]. *H. pylori* alters mitochondrial dynamics via the secreted toxin, vacuolating cytotoxin A (VacA) [223]. VacA induces mitochondrial localization and activation of DRP1, resulting in mitochondrial fragmentation [223]. DRP1 in mitochondria impairs the mitochondrial ETC as a result of loss of MMP, leading to mtROS production [224]. VacA also suppresses activation of mammalian target of rapamycin complex 1 (mTORC1) signaling [225]. The inhibition of mTORC1 can induce activation of DRP1 [226]. Therefore, *H. pylori* infection may regulate ROS production by altering mitochondrial dynamics.

The SipB toxin of the intracellular bacterium *Salmonella enterica* distorts mitochondrial cristae morphology in macrophages [227]. *B. abortus* and *B. melitensis* increase mitochondrial fragmentation in macrophages [228]. *L. monocytogenes* modulates mitochondrial dynamics, leading to mitochondrial fragmentation [229]. Listeriolysin O, a secreted pore-forming toxin of *L. monocytogenes*, increases Ca^2+^ influx into mitochondria, resulting in reduction of MMP, mitochondrial respiratory activity and ATP production [229]. Listeriolysin O eliminates damaged mitochondria by mitophagy, leading to suppression of ROS, promoting the survival of *L. monocytogenes* [230]. *Legionella pneumophila* is the causative agent of Legionnaires’ disease and has a type 4 secretion system (T4SS) for injection of bacterial effector proteins into host cells [231]. By recruiting the mitochondrial fission protein DRP1 [231], the T4SS effector protein MitF induces mitochondrial fission, resulting in removal of mitochondria and reducing the mtROS level [231]. Another intracellular bacterium, *Chlamydia trachomatis* also alters mitochondrial dynamics. [232]. *C. trachomatis*, a cause of sexually transmitted diseases, elevates intracellular cAMP and ROS levels, leading to phosphorylation of DRP1 serine residue 637 (S637) [232]. During infection, phosphorylated DRP1 S637 plays a key role in mitochondrial fission and suppression of *C. trachomatis* growth [232].

Mtb affects the mitochondrial network [233,234,235]. Mtb alters macrophage mitochondrial morphology from tubular to spherical or ovoid [233]. The spherical and ovoid mitochondria are dysfunctional, exhibiting loss of MMP and reduction of cytosolic ATP and upregulation of mtROS production [233]. The attenuated Mtb strain, H37Ra, results in increased swelling of mitochondria compared to the virulent H37Rv strain in macrophages [234]. In addition, Mtb H37Rv more strongly induced the MMP than Mtb H37Ra [234]. In addition, Mtb changes the mitochondrial fusion–fission balance, leading to mitochondrial fragmentation [235]. Mtb H37Ra degrades the mitochondrial fusion protein MFN2 in macrophages, resulting in increased mitochondrial fragmentation [235]. ROS induce localization of the E3 ligase Parkin in mitochondria [236]. In Mtb-infected macrophages, Parkin promotes mitochondrial fragmentation by degrading MFN2 [235]. Therefore, reciprocal regulation of ROS is likely to be linked to mitochondrial dynamics during mycobacterial infection.

In summary, mitochondrial fragmentation and dysfunction are important for regulation of pathogens. However, the role of mtROS in pathogen elimination is unclear. Further studies are needed to determine the link between mitochondrial dynamics and pathogen elimination.

## 6. Conclusions

ROS generation activates antimicrobial mechanisms in host cells, suppressing microbial growth and survival. The effects of ROS may be beneficial or detrimental, depending on the quantity. ROS directly target diverse microbes, or induce autophagy and/or apoptosis, leading to pathogen elimination. By contrast, excess ROS cause oxidative damage and ultimately cell death, inducing inflammation and leading to systemic damage. However, how pathogens regulate oxidative stress in immune cells is unknown, as is the relative importance of mtROS and cytosolic ROS in pathogen elimination. Understanding the regulatory mechanisms of ROS production and how ROS modulate immune-cell function to remove pathogens will facilitate the development of novel therapeutics for intractable infectious diseases.

## Figures and Tables

**Figure 1 antioxidants-10-00872-f001:**
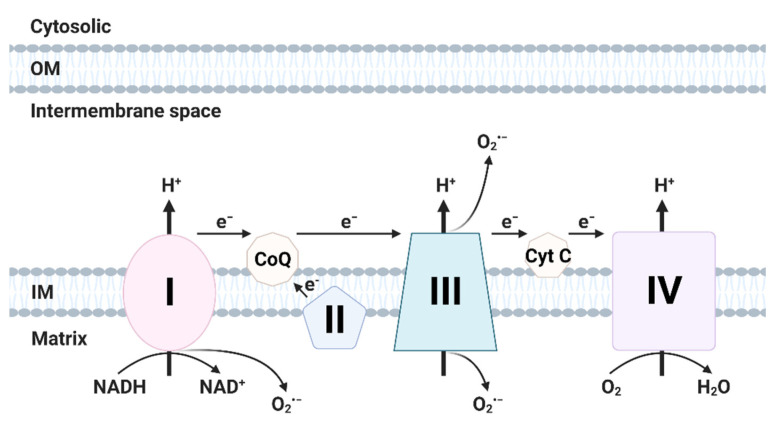
Generation of ROS in electron transport chain (ETC). The ETC is located in the mitochondrial inner membrane (IM). Complex I and II supply electrons to coenzyme Q (CoQ; ubiquinone). Sequentially, electrons are transferred from CoQ to Complex III, cytochrome c (Cyt c) and Complex IV. Oxidative stress is generated during electron transfer. (Created with BioRender.com accessed on 29 March 2021).

**Figure 2 antioxidants-10-00872-f002:**
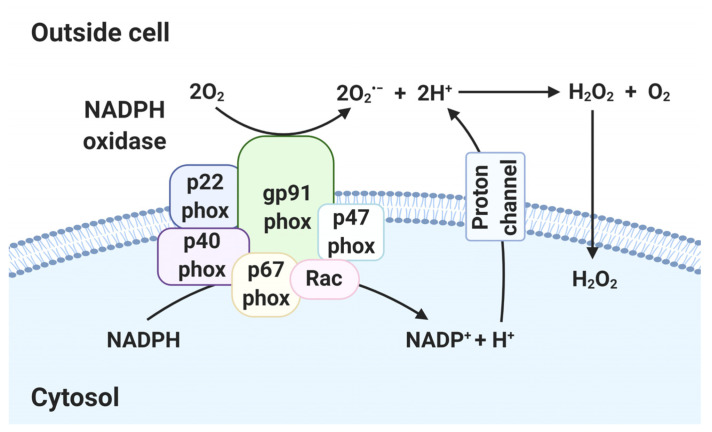
Generation of ROS in NADPH oxidase (NOX). In NOX, electrons of cytoplasmic NADPH are transferred to extracellular oxygen via gp91phox, forming O_2_^•−^. Other subunits (p22phox, p40phox, p47phox, p67phox and Rac) are responsible for NOX stabilization and regulation. The proton channel transfers protons extracellularly. O_2_^•−^ is converted to H_2_O_2_ and diffuses through lipid membranes into the intracellular space. (Created with BioRender.com accessed on 29 March 2021).

**Figure 3 antioxidants-10-00872-f003:**
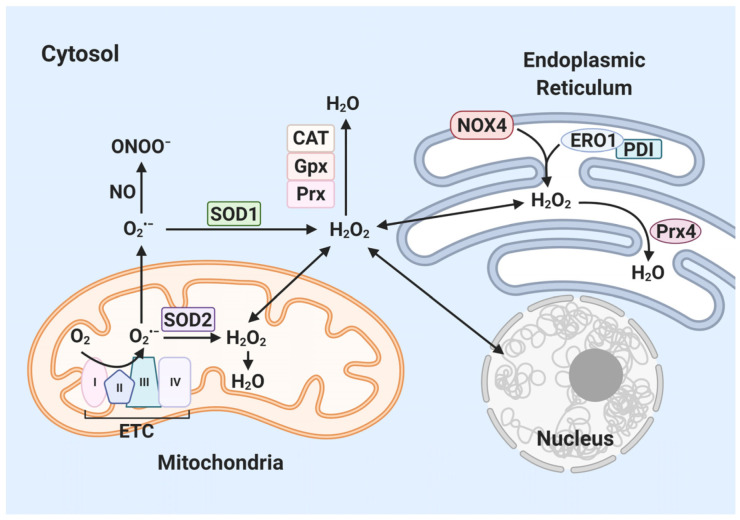
Generation of ROS in cytosol. Superoxide (O_2_^•−^, precursor of ROS) is generated in mitochondria and endoplasmic reticulum. O_2_^•−^ produced in the mitochondrial ETC is dismutated into H_2_O_2_ by superoxide dismutase (SOD) 1/2. H_2_O_2_ is converted to water by catalase (CAT), glutathione peroxidases (Gpx) and peroxiredoxins (Prx). O_2_^−^ reacts with NO to produce ONOO^−^. Inside the ER, ER oxidoreductin 1 (ERO1)/protein disulfide isomerase (PDI) and NADPH oxidase 4 (NOX4) produce H_2_O_2_. Prx4 reduces H_2_O_2_ to water via a catalytic cysteine residue. (Created with BioRender.com accessed on 29 March 2021).

**Figure 4 antioxidants-10-00872-f004:**
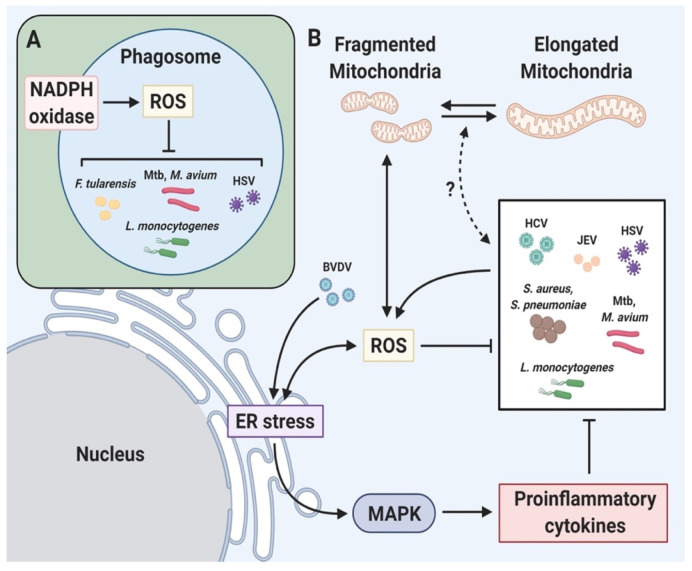
Relationship between ROS and pathogens. Pathogen elimination via ROS-mediated immune signaling. (**A**) Pathogens are recognized by PRRs, phagocytized and digested. NOX is assembled and activated to produce ROS in the phagosomal membrane, resulting in elimination of the pathogen. (**B**) ER stress and mitochondrial fragmentation induce ROS production. ROS-induced ER stress boosts proinflammatory cytokine production via MAPK, resulting in microbial killing. Illustration of pathogen-mediated regulation of immune signaling via ROS production. HCV, hepatitis C virus; JEV, Japanese encephalitis virus; HSV, herpes simplex virus 1; BVDV, bovine viral diarrhea virus; *S, aureus*, *Staphylococcus aureus*; *S. pneumoniae Streptococcus pneumoniae*; *M. tuberculosis*, *Mycobacterium tuberculosis*; *M. avium, Mycobacterium avium*; *L. monocytogenes, Listeria monocytogenes*; *F. tularensis, Francisella tularensis.* (Created with BioRender.com accessed on 29 March 2021).

**Figure 5 antioxidants-10-00872-f005:**
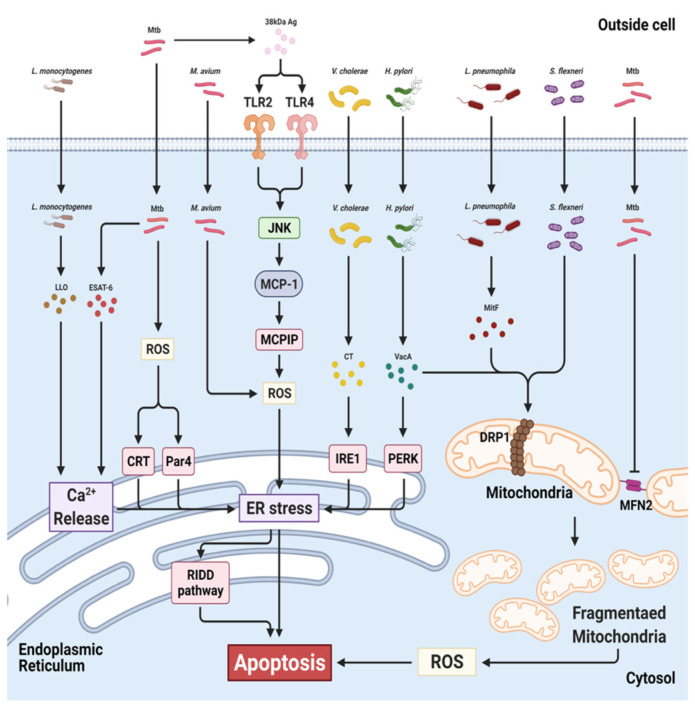
Targets of bacteria in ER and mitochondria during infection. Mtb induces ER stress-mediated apoptosis via ROS, calreticulin (CRT), Par-4 (Par4) and Ca^2+^ release. *M. avium* increases ROS-mediated ER stress, leading to activation of the RIDD pathway. Cholera toxin (CT) of *V. cholera* phosphorylates IRE1. *Helicobacter pylori* secretes VacA, leading to activation of PERK and DRP1. *L pneumophila* and *S. flexneri* induce mitochondrial fragmentation in a DRP1-dependent manner. Mtb also triggers mitochondrial fragmentation by inhibiting MFN2. Mtb, *Mycobacterium tuberculosis*; *M. avium, Mycobacterium avium*; *L. monocytogenes, Listeria monocytogenes*; *H. pylori, Helicobacter pylori*; *V. cholera, Vibrio cholera*; *L pneumophila, Legionella pneumophila*; *S. flexneri, Shigella flexneri.* (Created with BioRender.com accessed on 29 March 2021).

## Data Availability

Data available in a publicly accessible repository.

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
