# Peer review of "Effect of Reactive Oxygen Species on the Endoplasmic Reticulum and Mitochondria during Intracellular Pathogen Infection of Mammalian Cells"

_antioxidants, 2021, doi:10.3390/antiox10060872_

Round 1

Reviewer 1 Report

This is a review of ROS and cellular mechanisms involving them, specifically in mitochondria and the ER, with an emphasis on these in cells of various types exposed to a variety of pathogens and other treatments. The figures are informative and appear accurate. While the discussion of individual enzyme systems and pathways seems well organized and likely accurate, the subsection topics don't appear to follow a logical order, and could be better integrated with one another.
The major problem with this review is the language. Virtually every paragraph contains errors in grammar or word usage such that the intended meaning is unclear. The impression is the authors have a good understanding of the reviewed material; however, the review requires reorganization and rewriting to make  this understanding coherent and clear to the reader.
The title itself is misleading: 'infections' if not otherwise qualified is normally understood to mean 'infectious diseases' yet this review deals instead with CELLULAR  infections, and no attempt is made to connect the in vitro cellular mechanisms described to whole-organism disease processes. Moreover, in some of the experiments discussed, cells are not treated with pathogens, hence 'during pathogen infections' is somewhat inaccurate.
The definition of ROS (line 36) is wrong: not all ROS are free radicals nor do they typically contain oxygen ions (as distinct from oxygen atoms). ROS are of such great importance in this review that this otherwise trivial error cannot be dismissed.
By and large the cell types and the inducing treatments used in the experiments cited are not disclosed. This gives the impression that the pathways and mechanisms apply to all kinds of cells without regard to treatment. In some cases, the cell types (adipocytes, neuroblastoma cells) have nothing to do with infections.
The review appears a bit dated, with few references from 2018 on. Somewhat surprisingly, SARS-2 is not among the pathogens discussed.

Author Response

Reviewer 1

This is a review of ROS and cellular mechanisms involving them, specifically in mitochondria and the ER, with an emphasis on these in cells of various types exposed to a variety of pathogens and other treatments. The figures are informative and appear accurate. While the discussion of individual enzyme systems and pathways seems well organized and likely accurate, the subsection topics don't appear to follow a logical order, and could be better integrated with one another.

The major problem with this review is the language. Virtually every paragraph contains errors in grammar or word usage such that the intended meaning is unclear. The impression is the authors have a good understanding of the reviewed material; however, the review requires reorganization and rewriting to make this understanding coherent and clear to the reader.

  1. R) Thank you for your valuable comments. We revised it. Our manuscript was checked by native speakers of English as follows: The English in this document has been checked by at least two professional editors, both native speakers of English. For a certificate, please see: http://www.textcheck.com/certificate/lkBU1B

The title itself is misleading: 'infections' if not otherwise qualified is normally understood to mean 'infectious diseases' yet this review deals instead with CELLULAR infections, and no attempt is made to connect the in vitro cellular mechanisms described to whole-organism disease processes. Moreover, in some of the experiments discussed, cells are not treated with pathogens, hence 'during pathogen infections' is somewhat inaccurate.

  1. R) Thank you so much for your comments. To be clarified, we revised the title as follows; “Effect of Reactive Oxygen Species on the Endoplasmic Reticulum and Mitochondria during Intracellular Pathogen Infection of Mammalian Cells”

The definition of ROS (line 36) is wrong: not all ROS are free radicals nor do they typically contain oxygen ions (as distinct from oxygen atoms). ROS are of such great importance in this review that this otherwise trivial error cannot be dismissed.

  1. R) In order to clarify it, we deleted the text “ROS are free radicals which are containing oxygen ions. Reactivity of free radicals is high because existing unpaired electrons.” in “2.1. ROS in mitochondria” section.

By and large the cell types and the inducing treatments used in the experiments cited are not disclosed. This gives the impression that the pathways and mechanisms apply to all kinds of cells without regard to treatment. In some cases, the cell types (adipocytes, neuroblastoma cells) have nothing to do with infections.

  1. R) To be clarified, we revised it. We focused on immune cells according to your comments.

The review appears a bit dated, with few references from 2018 on. Somewhat surprisingly, SARS-2 is not among the pathogens discussed.

  1. R) We changed or added the references for proper expression. In addition, information on SARS-CoV-2 has been added in “3.3. ROS in viral pathogenesis” section as follows; “Severe acute respiratory syndrome coronavirus-2 (SARS-CoV-2) induces sepsis in severe cases. SARS-CoV-2 sepsis induces hypoxia, which can promote mitochondrial dysfunction leading to production of superoxide, H2O2, and other ROS. In addition, virus-induced ROS damage the erythrocyte membrane, and damaged erythrocytes are phagocytosed by macrophages and neutrophils, further increasing ROS production. The interplay between excess ROS and cytokines induces a cytokine storm and oxidative stress production, leading to death from sepsis and shock. Cytokine storm causes acute respiratory distress syndrome and multiple organ dysfunction in patients with SARS-CoV-2. Therefore, regulation of ROS could have therapeutic potential for SARS-CoV-2 infection.”

Reviewer 2 Report

This is an exciting revision on the ROS production during bacterial and viral infections and its innate immunity role. I have only a few minor comments:

-The abstract is very short, and the authors should develop this part of the manuscript further. Besides, the authors should modify the last sentence as it is not appropriate in this context.

-The English language should be proofread by a native speaker. Some parts of the text are difficult to follow.

-It is quite surprising that in the introduction, the authors do not mention the oxidative burst during phagocytosis since this is the main mechanism to eradicate pathogens, and it is based on ROS production.

-Salmonella Typhimurium is not properly written.

-Finally, the authors do not mention the role of reactive nitrogen species in this context. Some information on RNS would greatly improve the review if the authors covered this point.

Author Response

Reviewer 2

This is an exciting revision on the ROS production during bacterial and viral infections and its innate immunity role. I have only a few minor comments:

-The abstract is very short, and the authors should develop this part of the manuscript further. Besides, the authors should modify the last sentence as it is not appropriate in this context.

  1. R) Thank you for your advice. We have reinforced the content of the abstract.

-The English language should be proofread by a native speaker. Some parts of the text are difficult to follow.

  1. R) We revised it. Our manuscript was checked by native speakers of English as follows: The English in this document has been checked by at least two professional editors, both native speakers of English. For a certificate, please see: http://www.textcheck.com/certificate/lkBU1B

-It is quite surprising that in the introduction, the authors do not mention the oxidative burst during phagocytosis since this is the main mechanism to eradicate pathogens, and it is based on ROS production.

  1. R) Thank you for your valuable comments. We have added the text the oxidative burst in introduction as follows; “Phagocytized pathogens are degraded in phagocytes, which is boosted by the so-called oxidative burst (or respiratory burst). The oxidative burst is an important component of immunity and removing microbes. After infection, rapidly occurring oxidative burst is involved in the generation of nitric oxide and reactive oxygen species (ROS).”

-Salmonella Typhimurium is not properly written.

  1. R) We revised it in “3.2. ROS in bacterial pathogenensis” section as follows; “Mitochondrial-targeted catalase expression inhibited the killing of Salmonella typhimurium, demonstrating that mitochondrial ROS can be bactericidal.”

-Finally, the authors do not mention the role of reactive nitrogen species in this context. Some information on RNS would greatly improve the review if the authors covered this point.

  1. R) We have added the text about RNS in introduction as follows; “Nitric oxide is generated by nitric oxide synthases (NOS), which convert L-arginine into L-citrulline and nitric oxide. There are three NOS isoforms, one of which, inducible-NOS (iNOS), is induced in various cell types by cytokines and other agents in response to infection and inflammation. Mitochondrial NOS-derived nitric oxide produces oxidative nitrogen oxide species, resulting in oxidative damage to mitochondria and apoptosis. Several cellular signaling functions of nitric oxide are executed in mitochondria. Nitric oxide inhibits activation of cytochrome c oxidase by competing with O2 for binding to cytochrome c oxidoreductase. Nitric oxide reacts with complex III of the mitochondrial electron transport chain (ETC) to inhibit electron transfer and induce O2 generation. Therefore, nitric oxide is involved in the regulation of host mitochondrial and cellular functions.”

Reviewer 3 Report

Dear authors,

Presented for review manuscript entitled: "The effect of oxidative stress on ER and mitochondria during pathogen infections" is well written and shows review of literature about oxidative stress effect on mitochondria and ER during pathogen infection from last 20 years. In this review authors cited literature from last 20 years and around half of reviewed literature was published during last 10 years. I have a few minor comments:

  1. I propose to write a paragraph more general paragraph at the beginning of manuscript in which authors describe other ROS and their genesis.
  2. In subsection 3.3 it would be nice to mention about the ROS during Coronaviruses infection or during SARS-CoV-2 infection.
  3. 2 ways to do. First more precise title that this review concerns the effect of oxidative stress on mitochondria and ER during pathogens infection in mammals or animals or human. Or second way add section about effect of oxidative stress on mitochondria and ER during pathogens infection in plants, because there is a lot of data showing the connection of oxidative stress effects on mitochondria and ER in plants during pathogen infection.
  4. Bibliography: the position 15: the year of publication should be bolded.

Generally I accept this paper for publication in antioxidants after minor revision.

Author Response

Reviewer 3

Dear authors,

Presented for review manuscript entitled: "The effect of oxidative stress on ER and mitochondria during pathogen infections" is well written and shows review of literature about oxidative stress effect on mitochondria and ER during pathogen infection from last 20 years. In this review authors cited literature from last 20 years and around half of reviewed literature was published during last 10 years. I have a few minor comments:

  1. I propose to write a paragraph more general paragraph at the beginning of manuscript in which authors describe other ROS and their genesis.
  2. R) We revised it.
  3. In subsection 3.3 it would be nice to mention about the ROS during Coronaviruses infection or during SARS-CoV-2 infection.
  4. R) Information on SARS-CoV-2 has been added in “3.3. ROS in viral pathogenesis” section as follows; “Severe acute respiratory syndrome coronavirus-2 (SARS-CoV-2) induces sepsis in severe cases. SARS-CoV-2 sepsis induces hypoxia, which can promote mitochondrial dysfunction leading to production of superoxide, H2O2, and other ROS. In addition, virus-induced ROS damage the erythrocyte membrane, and damaged erythrocytes are phagocytosed by macrophages and neutrophils, further increasing ROS production. The interplay between excess ROS and cytokines induces a cytokine storm and oxidative stress production, leading to death from sepsis and shock. Cytokine storm causes acute respiratory distress syndrome and multiple organ dysfunction in patients with SARS-CoV-2. Therefore, regulation of ROS could have therapeutic potential for SARS-CoV-2 infection.”
  5. 2 ways to do. First more precise title that this review concerns the effect of oxidative stress on mitochondria and ER during pathogens infection in mammals or animals or human. Or second way add section about effect of oxidative stress on mitochondria and ER during pathogens infection in plants, because there is a lot of data showing the connection of oxidative stress effects on mitochondria and ER in plants during pathogen infection.
  6. R) Thank you for your comment. To be clarified, we revised the title as follows; “Effect of Reactive Oxygen Species on the Endoplasmic Reticulum and Mitochondria during Intracellular Pathogen Infection of Mammalian Cells”
  7. Bibliography: the position 15: the year of publication should be bolded.
  8. R) We have corrected for reference.

Generally I accept this paper for publication in antioxidants after minor revision.

Round 2

Reviewer 1 Report

This is the revised version of a review of ROS and cellular mechanisms involving them, specifically in mitochondria and the ER, with an emphasis on these in cells of various types exposed to a variety of pathogens and other treatments. The concerns raised with respect to the original version have been satisfied in the main.
The following minor points needing correction were noted:
(1) Lines 66-67 'five subunits'...'complexes I-IV'. The number of subunits and complexes should agree.
(2) Line 82. Use of 'alters' is not correct here. The sentence could be corrected by replacing 'alters reduced' by 'is reduced to'.
(3) Line 89. Use of 'reaction' is incorrect and the sentence requires revision.
(4) Molecular formulas in the Figure 2 and Figure 3 legends and on line 319 need to be subscripted appropriately.
(5) The sentence on lines 218-220 needs to be revised to make clear that it is the experimental treatment with ginseng that is the source of NOX inhibition, not the H. pylori infection.

Author Response

Author’s response

Reviewer 1

This is the revised version of a review of ROS and cellular mechanisms involving them, specifically in mitochondria and the ER, with an emphasis on these in cells of various types exposed to a variety of pathogens and other treatments. The concerns raised with respect to the original version have been satisfied in the main.

The following minor points needing correction were noted:

(1) Lines 66-67 'five subunits'...'complexes I-IV'. The number of subunits and complexes should agree.

A) We revised it as follows; “The mitochondrial respiratory chain comprises four subunits (respiratory complexes I–IV)”

(2) Line 82. Use of 'alters' is not correct here. The sentence could be corrected by replacing 'alters reduced' by 'is reduced to'.

A) We revised it according to your comment; “Coenzyme Q is reduced to ubiquinol (reduced form of coenzyme Q) by receiving two electrons from mitochondrial complex I or II.”

(3) Line 89. Use of 'reaction' is incorrect and the sentence requires revision.

A) We revised it as follows; Superoxide is produced by electron of ubisemiquinone in complex III and released into the intermembrane space [30].

(4) Molecular formulas in the Figure 2 and Figure 3 legends and on line 319 need to be subscripted appropriately.

A) We revised it.

(5) The sentence on lines 218-220 needs to be revised to make clear that it is the experimental treatment with ginseng that is the source of NOX inhibition, not the H. pylori infection.

A) We revised the text as follow; During Helicobacter pylori infection, inhibition of NOX by Korean red ginseng extract deactivates the Janus kinase (JAK) 2/ signal transducer and activator of transcription (STAT) 3 pathway, downregulating MCP1 [93].